# Using a Dynamic Model to Estimate the Cost-Effectiveness of HPV Vaccination in Iran

**DOI:** 10.3390/vaccines12040438

**Published:** 2024-04-18

**Authors:** Arnold Hagens, Albertus Constantijn Sloof, Roksana Janghorban

**Affiliations:** 1Department of Health Sciences, University Medical Center Groningen, University of Groningen (RUG), 9713 AV Groningen, The Netherlands; 2Triangulate Health Ltd., Doncaster DN11 9QU, UK; 3Asc Academics B.V., 9725 AK Groningen, The Netherlands; 4Maternal-Fetal Medicine Research Center, Department of Midwifery, School of Nursing and Midwifery, Shiraz University of Medical Sciences, Shiraz 71936-13119, Iran; janghorban@sums.ac.ir

**Keywords:** human papillomavirus, vaccine, cost-effectiveness, low- and middle-income countries, cervical cancer, quality-adjusted life years

## Abstract

This study aimed to determine the cost-effectiveness of vaccination against HPV. An age–sex structured dynamic disease transmission model was created to estimate the spread of HPV and the HPV-related incidence of cervical cancer (CC) in Iran. Sixteen age groups of men and women were incorporated to reflect the differences in sexual preferences, vaccination uptake, and disease-related outcomes. Three scenarios were evaluated by using an Incremental Cost-Effectiveness Ratio (ICER) with gained quality-adjusted life years (QALYs). ICER values below one gross domestic product (GDP) per capita are evaluated as highly cost-effective. Vaccination reduces the number of infections and CC-related mortality. Over time, the vaccinated group ages and older age groups experience protection. An initial investment is required and savings in treatment spending reduce the impact over time. Vaccinating girls only was found to be cost-effective, with an ICER close to once the GDP per capita. Vaccinating both sexes was shown to be less cost-effective compared to girls only, and vaccinating boys only was not found to be cost-effective, with an ICER between once and three times, and greater than three times the GDP per capita, respectively. The estimates are conservative since societal cost-saving and the impact of other HPV-related illnesses were not considered and would likely reduce the ICERs.

## 1. Introduction

Genital human papillomavirus (HPV) infection is the most common sexually transmitted infection (STI) and is usually passed from one person to another after the first sexual activity [1]. According to the World Health Organization (WHO), in 2019, about 600 million men and women worldwide were infected with the virus [2].

Persistent infection with HPV can lead to precancerous and cancerous lesions. Malignancies caused by HPV occur in the anus, cervix, oropharynx, penis, rectum, vagina, and vulva [3]. Both sex-related and regional disparities add to the burden and impact of HPV-associated malignancies. For example, the burden of HPV-related cancers is higher among women than men. And black women show a high incidence rate of cervical cancer (CC) and have the highest CC mortality rate among all racial/ethnic groups [4]. 

Cervical cancer (CC) is preventable and curable in the early or precancerous stages. To mobilize efforts to eliminate cervical cancer, the WHO has set strategic targets for all countries by 2030, known as the 90-70-90 targets: 90% of girls to be fully vaccinated by age 15; 70% of women to be screened twice in a lifetime for CC (by age 35 and 45); and 90% of women diagnosed with cervical disease to receive treatment [5]. HPV vaccination is considered to be the most effective approach for preventing HPV infection and related diseases.

While HPV vaccination is likely to be a cost-effective public health investment in both low-income and high-income countries, nationwide vaccinations have largely been adopted in high- or upper-middle-income countries. In low- and middle-income countries (LMICs), the implementation of HPV vaccines has been primarily through small-scale pilot or demonstration projects, with significant financial and programmatic challenges identified as barriers to national scale-up [6]. Therefore, more studies and in different contexts are needed to decide on HPV programs [7]. 

Some authors have already worked on this. A study evaluated the cost-effectiveness of HPV vaccination among girls and young women in Mongolia, using a discount rate of 3% and aggregated into an Incremental Cost-Effectiveness Ratio (ICER). The results showed that HPV vaccination in girls in Mongolia has a high probability of being a low-cost investment compared to no vaccination, with projected ICERs representing less than 20% of gross domestic product (GDP) per capita (USD 3735 in 2018) [8]. Another study assessed the cost-effectiveness of HPV vaccination for adolescent girls and boys in the UK to prevent six HPV-caused cancers (cervical, anal, vaginal, vulvar, penile, and oropharyngeal). The results showed that vaccinating girls is cost-effective compared with no vaccination, while vaccinating both sexes is less cost-effective [9].

The Eastern Mediterranean Region (EMR) has experienced a concerning rise in the burden of cervical cancer, as evidenced by data on the absolute number of deaths and disability-adjusted life years (DALYs) from 2000 to 2017. Iran, within the EMR, displays a significant 82.7% relative increase in the number of deaths and an increase of 7936 DALYs during this period [10]. Moreover, cervical cancer has imposed a considerable economic burden, amounting to USD 206 million, positioning it as the second most financially demanding gynecologic cancer [11]. Based on the available evidence, no studies have yet undertaken a cost-effectiveness analysis of HPV vaccination in Iran. To address this gap, the present study was designed to conduct a cost-effectiveness analysis in Iran to evaluate the potential health and economic impact of HPV vaccination among women and men across potential introduction strategies.

## 2. Material and Methods

### 2.1. Overview

An age–sex structure deterministic model was used to simulate the Iranian population and to estimate the cost-effectiveness of various vaccination uptakes of men and women at certain ages. Vaccination scenarios were compared with a baseline of no vaccination. The time horizon was set at 70 years to give a more accurate estimate as certain effects occur at a later age. The period reflects roughly the expected years of life left of a 10-year-old man or woman in Iran [12]. The compartmental model was based on a SIRD (susceptible, infectious, recovered, and death) compartmental epidemiological model. A compartmental model is based on the concept of various health states where individuals move. To make the model fit with HPV and cervical cancer disease paths with vaccination, some adjustments were made. First, we included a vaccinated group containing individuals who are vaccinated against HPV and have certain immunity. Next, a differentiation was made between Infectious and Vaccinated Infectious because we assumed that it is possible for vaccinated individuals to still become infected and transmit the infection, but have a different progression towards CC. Next, a latency compartment was included which reflects the period between infection and actual CC development. An important aspect of the model is that we assume that not all CC developments are identified at an early stage. Therefore, two CC paths were included reflecting treated and untreated patients. From all these health states, Recovered or Death is possible. A schematic overview is shown in Figure 1. 

Sixteen age groups of men and women were incorporated in order to reflect differences between individuals of different ages and sex for various parameters, such as contacts, vaccine effectiveness, case-fatality rates, and vaccination uptake. To accurately simulate a longer time period, forecasted birthrate and background mortality rates were included in the model as well as estimated shifts between age groups [13]. Newborns were only added to the susceptible group, and we assumed background mortality to occur in every health state. 

Vaccinations were modeled by shifting individuals from the susceptible group to the vaccinated group. No immunity waning was included in the model as evidence shows a long-lasting protection against HPV [14]. Furthermore, post infection or -disease, both vaccinated and unvaccinated individuals move to Immune. 

All the simulations were carried out in R Version 4.2.1 by implementing the 16-age-2-sex-group compartmental model using the Runge–Kutta 4 method to approximate the solutions of the differential equations. We used Microsoft Excel to carry out additional analyses and tabulations. 

For the parametrization of the model we used, where possible, data from Iran. Details of all the parameters are included in Appendix A. Cost- and population-related parameters were taken from Iran. However, for certain epidemiological parameters, we depended on other sources as no specific country data for Iran were found. These included the contact matrix, the duration of infectious period, CC latency, vaccination effectiveness on infection, the fraction of cervical cancer patients treated, duration with cervical cancer treated, and the fraction of treated cervical cancer patients to die. We assumed that although they are from different settings, these epidemiological parameters do not deviate considerably.

### 2.2. Dynamic Transmission Model

The transmission of the HPV infection occurs through direct vaginal sexual relations [3]. In order to simulate this, a contact matrix has been constructed which is the core of the transition from the Susceptible and Vaccinated state to the Infectious states. As there is very limited research and evidence published on how mixing occurs, the matrix is used in combination with a calibrated Basic Reproduction number which is the expected number of cases directly caused by a single case in a population where all individuals are susceptible to infection. The calibration was based on prevalence rates in Iran [15]. We constructed the contact matrix by using the sexual activity per year [16], the average difference between men and women at the moment of the sexual relation [17], and an assumed standard deviation of the partners’ age. The resulting matrix was adjusted for the proportion of the groups in the population. Next, we obtained the eigenvalue of the contact matrix to calculate the common *β* from calibrated R_0_. The differential equations for the Infectious (1) and Vaccinated Infectious (2) states are:(1)$$\frac{dI_{i}}{dt}=\beta S_{i\;}\sum_{i}C_{ij}{(I}_{j}+{Iv}_{j})/N_{j}-\gamma \cdot I_{i}$$
(2)$$\frac{d{Iv}_{i}}{dt}={Vac\_Eff\_Inf}_{i}\cdot \beta S_{i}\sum_{i}C_{ij}{(I}_{j}+{Iv}_{j})/N_{j}-\gamma \cdot {Iv}_{i}$$
where *C_ij_* is the contact matrix over the age classes *i* with *j*, and *γ* represents the reciprocal of the infectious duration of the HPV infection. *β* is the infection rate according to the probability of infection in case of a contact. Additionally, we assumed that vaccinated individuals can still be infected and transmit the infection, and therefore they have been included in Equations (1) and (2). Nevertheless, the probability of infection of a vaccinated individual is reduced by the vaccine effectiveness on infection (*Vac_Eff_Inf_i_*). Utility values for estimating QALYs were estimated by interpolating existing utility curves from Iran [18]. The disutility for the CC states was estimated based on similar diseases in relation with the prevalence rates [19] of the different CC stages [11,20]. We used a rate of 3.5% to discount the QALYs [21].

### 2.3. Costs

Direct health care costs were used to calculate the costs in the model. We used the health care costs of treated and untreated persons in the CC states, which involved costs of hospitalization, medicaments, and CC treatments in Iran. Costs of vaccination consist of a two-dose quadrivalent vaccination and administration costs and was set at USD 152, which included an assumed USD 1 for administration [22]. Total health care treatment costs were estimated by dividing the total costs of CC treatment in Iran by the number of treated CC cases, and was set at USD 26,850 [22]. Costs were discounted with a rate of 3.5% [21].

### 2.4. Scenarios

To gain insight into the cost-effectiveness of HPV vaccination, we defined four scenarios where we varied the uptake of vaccination of men and women. The outcomes of these simulations, costs, QALYs, and resulting ICERs were tabulated for analysis. Additionally, we looked at the budget curves over the next 70 years for the costs which included vaccination and CC treatment costs.

We defined a baseline scenario and three plausible intervention scenarios methodologically based on intuitive scenarios [23,24]. The baseline scenario was based on a situation where no individuals are being vaccinated and the HPV virus can spread without any barrier. This scenario reflects the current situation in Iran. For intervention scenario 1, we assumed that boys and girls are eligible for vaccination and the uptake is 80% for both sexes. For intervention scenario 2, we assumed that only girls can be vaccinated with an uptake of 80%. And for scenario 3, we assumed only boys are being vaccinated with an uptake of 80%. Each of the scenarios was evaluated by using the ICER values with gained QALYs. A willingness-to-pay threshold (WPT) was used to establish the level of cost-effectiveness. ICER values below once the GDP per capita were evaluated as a very cost-effective scenario, and negative ICERs were ranked as cost-saving scenarios [25]. Additionally, we carried out a one-way sensitivity analysis on key parameters of the intervention scenarios in order to gain insight into the robustness of the used model.

## 3. Results

### 3.1. Initial Situation

Our simulation model requires an initial epidemiological situation. This means that for every health state and every age group, we needed to define an initial value. Therefore, a 70-year run-in simulation was used without HPV vaccination to estimate an initial state for all the health states of all age groups. We used the same initial state for all four scenarios.

### 3.2. Scenario Analysis

In this section, we will analyze the baseline and the four vaccination intervention scenarios. First, we will look at the health outcomes, followed by the impact the scenarios have on the health budget, and last, we will analyze the cost-effectiveness and the budget impact.

#### 3.2.1. Health Outcomes

Figure 2 shows the female mortality in the country due to HPV by age group and over time. The higher intensity of the tiles shows a higher mortality in Iran in the respective year.

The mortality among women in the baseline (Figure 2a) shows that without vaccination no change in mortality is observed over time. In the scenario where both boys and girls are vaccinated (Figure 2b), there is a reduction in mortality from approximately 20 years onwards. This effect is caused by the period between HPV infection and the manifestation of cervical cancer leading to death, first in the younger age groups but quickly also in the older age groups. A similar effect is visible for only vaccinating girls (Figure 2c). However, vaccinating only boys shows a very limited reduction in female mortality (Figure 2d).

The aggregated annual female mortality in the country is shown in Figure 3. It shows that in the 70-year period, the annual mortality reduces most in intervention 1, where boys as well as girls are vaccinated. Intervention 3, only boys, has only a limited impact on female mortality, and intervention 2, only girls, has a result that approaches the mortality of intervention 1. The baseline reduction in mortality is caused by the reducing birthrate in Iran, resulting in a change in the demographic situation, which in turn leads to a reduction in the sexually active population and population size overall.

#### 3.2.2. Budget Impact

One of the critical aspects of vaccination programs are the related public health expenditures. Different from treatment interventions which have rapid effects, vaccination programs have a delayed effect. This was also clearly visible in the health outcomes where the mortality reduction appeared over a long period. We can observe a similar effect for the public health expenditures. The results of the baseline and the intervention scenarios are shown in Figure 4.

Figure 4 illustrates that across all interventions, the health expenditures surpass those of the baseline situation. Among the interventions, vaccinating both boys and girls (intervention 1) incurs a higher annual cost compared to intervention 2, which focuses solely on girls, and intervention 3, which targets only boys. However, it is worth noting that over the course of a 70-year period, the cost differences diminish. This reduction can be attributed, in part, to the delayed health benefits of the vaccination. The figure clearly indicates that even for HPV vaccination, realizing savings in the treatment of CC requires patience and is a long-term investment.

#### 3.2.3. Cost-Effectiveness Analysis

We have seen a reduction in mortality due to CC and higher health care expenditure. In this section, we will analyze the cost-effectiveness of the three intervention scenarios.

Table 1 shows the summary of the simulation. It shows the discounted health care-related costs, the discounted QALYs, and the total deaths due to CC of the baseline and intervention scenarios. Additionally, it shows the incremental costs, QALYs, and mortality due to CC. These are the differences between the baseline and the respective intervention scenarios. The incremental costs and QALYs are used to calculate the ICER. The ICER shows the cost per gained QALY.

Between scenarios 2 and 3, vaccinating girls results in not only the highest gain in QALYs, and lives saved, but also in the lowest CC. The ICER of USD 4949 per QALY gained is just above a once the GDP per capita of USD 4091 (2021) [13], meaning that it is approaching very cost-effective. Scenario 3, vaccinating only boys, has a higher ICER, USD 15,529 per QALY gained, and is not cost-effective as it is above three times the GDP per capita threshold.

When looking at the ICER of scenario 1, vaccinating boys and girls, of USD 7916 per QALY, it is just below twice the GDP per capita threshold and could be considered for certain situations. However, here, we disregard this option as the ICER of scenario 1 relative to scenario 2 is USD 19,771 per QALY gained. This means that the incremental costs and effects from a policy change of only vaccinating girls compared to vaccinating boys and girls exceeds three times the GDP per capita.

#### 3.2.4. Sensitivity Analysis

A sensitivity analysis was carried on every scenario for the discount rate, vaccination effectiveness on infection, the cost of vaccination, vaccination uptake, and the cost per CC case treated. For the discount rate, a lower value of 0% and an upper value of 7% was used, and for the vaccination effectiveness, −10% for the lower value and 100% effectiveness for the upper one. For the other sensitivity parameters, we used a variation of +/−10% for upper and lower values. The results are shown in Figure 5.

The order and the magnitude of sensitivity are similar for the three scenarios. The discount rate and the vaccination effectiveness on infection have the highest effect on the ICER. An increase in discount rate to 7% increases the ICERs of all three scenarios by 116.4%, 106.9%, and 87.9%, respectively. A decrease to 0% leads to an increase in the ICERs with 54.9%, 52.1%, and 47.7%, respectively. An increase in the vaccination effectiveness leads to a decrease in the ICERs with 7.3%, 4.9%, and 7%, respectively. A decrease leads to an increase of the ICERs by 86.5%, 65.8%, and 72.3%, respectively. The cost of vaccination is less sensitive and an increase leads to an increase in the ICERs by 11.3%, 10.8%, and 10.4%, respectively. A decrease leads to a decrease in the ICERs by 11.3%, 10.8%, and 10.4%, respectively. The cost per CC case treated and the vaccination uptake have considerably lower sensitivity and minimal impact on the ICERs.

## 4. Discussion

The results of this analysis show that vaccinating against HPV infection is cost-effective in Iran when vaccinating only girls. Vaccinating only boys is less cost-effective, and the incremental effects of vaccinating boys and girls exceed the cost-effectiveness threshold.

When vaccinating only girls, the ICER of USD 4949 per gained QALY is close to the once the GDP per capita of USD 4091. The simulation shows that in total 20,527 lives are saved over a period of 70 years and 170,782 QALYs are gained. The finding confirmed with the WHO’s definition of cost-effectiveness categorizes interventions as highly cost-effective, cost-effective, or not cost-effective based on their ICER being less than one, between one and three, or greater than three times the GDP per capita [26]. Although an additional 4968 lives and 42,734 QALYs can be gained when vaccinating boys and girls, the corresponding ICER of USD 15,529 per gained QALY exceeds three times the GDP per capita threshold. However, it could be argued that it still is approaching the threshold of three times the GDP per capita. and with the development of the GDP in Iran, this intervention could be considered in the near future. A meta-regression analysis on the cost-effectiveness of HPV vaccination for both sex in 195 countries showed that the adjusted mean predicted ICER is 2017 USD 9222 per DALY averted (95% UI): USD 1683–28,936 in Iran [27].

The cost-effectiveness of HPV vaccination to prevent cervical cancer has been widely studied in multiple settings and with multiple vaccination strategies. However, to the best of our knowledge, this is the first cost-effectiveness study using a dynamics disease transmission model in Iran. Due to the wide ranges in settings and vaccination strategies used in the literature, it is difficult to compare our results to previously obtained results. For example, a cost-effectiveness analysis of HPV vaccination implementation in four countries (India, Vietnam, Uganda, and Nigeria) found costs per DALY ranging from USD 93 to USD 1406 [28]. Our results can be considered to be conservative when compared to the literature of cost-effectiveness studies in LMICs [28,29,30]. Additionally, in countries similar to Iran, HPV vaccination in girls only was found to be cost-effective. In Mongolia, an ICER less than 20% of GDP per capita was found. In China, vaccination only was shown to save 632 QALYs in a cohort of 100,000 girls using a lifetime horizon. Using a low cost of vaccination of USD 50 per two doses resulted in a highly cost-effective intervention of screening and vaccination versus screening only. Including productivity losses using a societal perspective has the possibility to reduce the ICER by more than 50% in certain cases [7]. Certain studies also included the prevention of genital warts or multi-cancer prevention as additional benefits of the vaccination, which might have resulted in lower cost-effectiveness outcomes [29,31,32,33,34,35].

The estimated impact on the government budget is expected to range between USD 50 and 100 million per year, which accounts for approximately 0.015% of Iran’s total GDP. This calculation takes into consideration both the additional costs incurred and the savings achieved in health care expenditures. It is important to note that these figures could be partially offset by reduced health care expenses related to other HPV-related diseases, decreased productivity losses, and an increase in HPV-related tax revenue. Additionally, as time progresses, the overall treatment costs of cervical cancer are projected to decrease, leading to a reduction in the required government budget.

The sensitivity analysis on the ICERs shows that there are no considerable differences in order and sensitivity between the three scenarios. However, the high sensitivity of the discount rate could considerably affect the ICERs, doubling or halving them. The sensitivity of the vaccination effectiveness could lead to a 65.8–86.5% increase in the ICER when the effectiveness is reduced by 10%. Such effects should be considered in future research and policy making. The sensitivity of the cost of vaccination, cost per CC case treated, and vaccination uptake show rather low and robust effects.

The potential for scaling up HPV vaccination in Iran is not only attractive but also feasible. Several key factors support this proposition. Firstly, HPV vaccination has been successfully implemented in numerous countries, providing strong evidence of its effectiveness. Secondly, research conducted on this topic has consistently demonstrated that vaccinating girls against HPV is cost-effective. Furthermore, considering the positive outcomes and potential benefits, a next step could extend the vaccination program to include boys too. Finally, while the budget impact of HPV vaccination is expected to gradually decrease over time, there is a strong likelihood of compensation through reduced health care costs and other related advantages. The implementation of female adolescent HPV vaccination in some countries like Rwanda confirmed the benefits. The country achieved a successful short-term impact, a vaccination coverage of above 90%, and long-term outcomes, decreasing the age standardized incidence of CC from 42 in 2010 to 28.2 in 2020 per 100,000 females.

This research has several strengths and limitations. The strength of this research is that we elaborated a mathematical model that works with disaggregated age and sex data that could be used as a standard for other cost-effectiveness analyses related to infection disease interventions. This research has limitations. First, the sexual relations matrix used to drive the infections was conservatively estimated with data from other countries and the authors’ assumptions. Although the behavior of the matrix was calibrated with real data, the real dynamics could be different. Second, we did not focus on other cancers that could be caused by HPV or societal costs; their inclusion could affect the ICER and cost-savings due to the prevention of other types of cancers. Third, we did not include sexual contacts of the same sex because of limited data on sexual relations. Fourth, our focus was on direct health care costs, and indirect costs such as productivity loss due to absenteeism and presenteeism and premature death were not included. Future research could concentrate on these aspects to further improve the model and the estimates of cost-effectiveness for other infectious disease and vaccination programs. The model could be expanded with more health status compartments, reflecting other HPV-related diseases, and in addition, indirect/societal costs could be included to estimate the ICERs from a societal perspective.

We can conclude that the estimation of the cost-effectiveness and budget impact of introducing HPV vaccination in Iran is highly dependent on specific data, such as data on sexual mixing patterns and behaviors. Findings show that vaccinating girls will reduce the mortality rate over the coming 70 years in a cost-effective manner. This investment is likely highly cost-effective and has the potential for cost-saving when productivity gains and health expenditure impacts regarding other HPV-related cancers are included.

## Figures and Tables

**Figure 1 vaccines-12-00438-f001:**
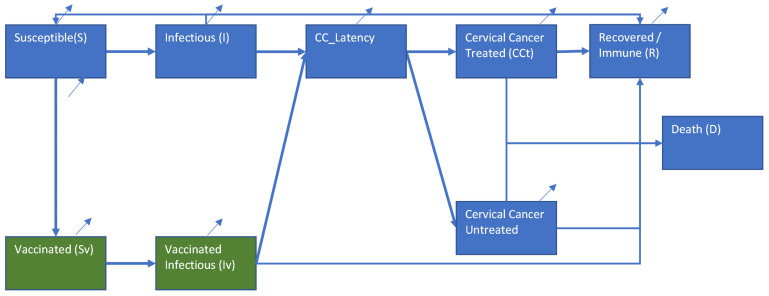
Schematic overview of the used HPV-CC model (small diagonal arrows represent background mortality and newborns).

**Figure 2 vaccines-12-00438-f002:**
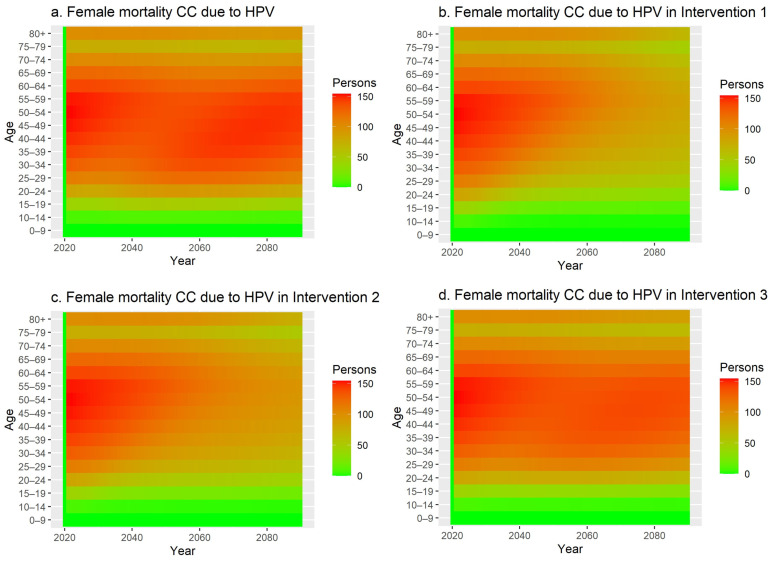
(**a**–**d**) Female mortality due to cervical cancer in the baseline scenario. ((**a**) top left) is the baseline scenario, ((**b**) top right) is intervention scenario 1 where boys and girls are vaccinated against HPV, ((**c**) bottom left) is intervention scenario 2 where only girls are vaccinated against HPV, and ((**d**) bottom right) is intervention scenario 3 where only boys are being vaccinated.

**Figure 3 vaccines-12-00438-f003:**
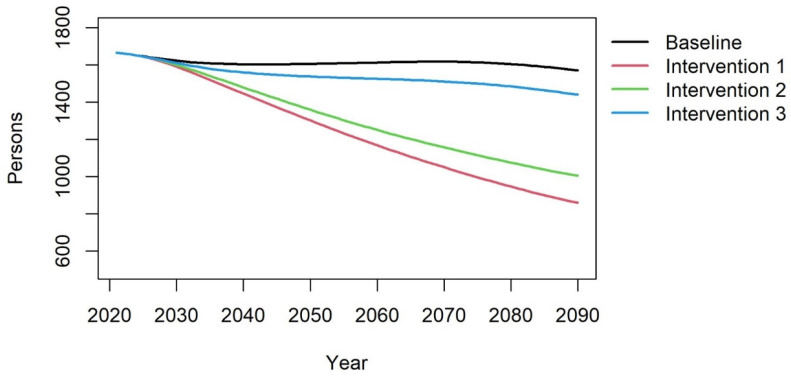
The female annual mortality of cervical cancer due to HPV infections for the baseline and the three intervention scenarios.

**Figure 4 vaccines-12-00438-f004:**
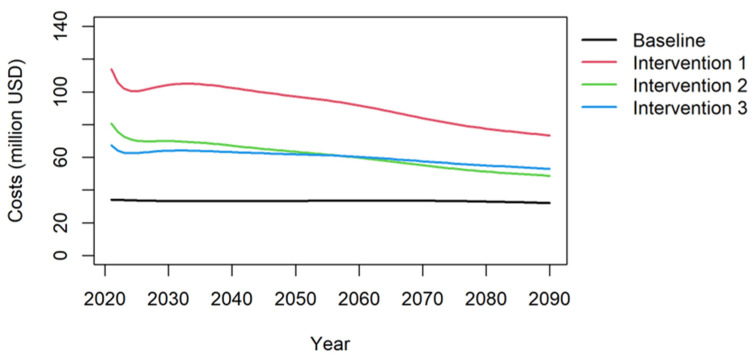
Annual public health expenditures of HPV vaccinations for the baseline and intervention scenarios.

**Figure 5 vaccines-12-00438-f005:**
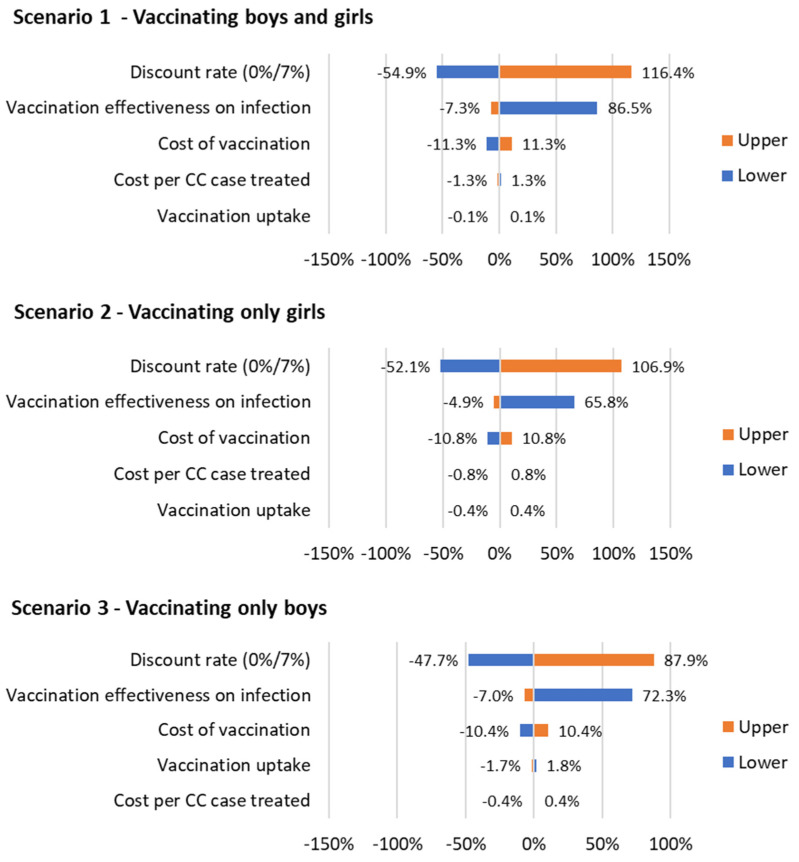
Sensitivity on the ICER of the three vaccination scenarios (+/−10% was used for the parameters, except for the discount rate (0%, 7%) and vaccination effectiveness on infection (−10% and 100%, being the max effectiveness)).

**Table 1 vaccines-12-00438-t001:** Cost-effectiveness results of the 70-year simulation.

Scenario	Costs (USD)	QALYs	Deaths	Incremental			ICER
				Costs (USD)	QALYs	Deaths	
Baseline (no vac)	867,751,387	2,158,770,017	112,796				
1. Vaccinating boys and girls	2,557,840,626	2,158,983,533	87,301	1,690,089,239	213,516	−25,496	7916
2. Vaccinating only girls	1,712,938,252	2,158,940,799	92,269	845,186,865	170,782	−20,527	4949
3. Vaccinating only boys	1,613,029,477	2,158,818,009	107,728	745,278,090	47,992	−5068	15,529

## Data Availability

The datasets used and/or analyzed during the current study are available from the corresponding author on reasonable request.

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
