# Peer review of "Using a Dynamic Model to Estimate the Cost-Effectiveness of HPV Vaccination in Iran"

_vaccines, 2024, doi:10.3390/vaccines12040438_

Round 1

Reviewer 1 Report

Comments and Suggestions for Authors

In the study, a mathematical model was elaborated that works with disaggregated age and sex data that could be used as a standard for cost-effectiveness analysis of interventions related with infection diseases. Vaccinating of girls against human papillomavirus was found to be cost-effective.

Notes:

1) line 17   This study aimed to determine the cost-effectiveness of the HPV

     Obviously the authors meant vaccination against HPV.

2) Line 191  Figure 2 shows the female mortality due to HPV

       It is necessary to clarify whether we are talking about mortality in the country or per 100,000 population

3) line 206 and 214   The aggregated annual female mortality is shown in Figure 3.

       See note to line 191

4) line 240 and 244  Additionally, it shows the incremental costs, QALYs and deaths.

       Probably means mortality from CC?

Author Response

Dear reviewer,

Thank you for your comments. We have attempted to address them a complete and suitable as possible. Below your questions and our responses. In addition the adjustments in the new submitted document are marked green.

Kind regards,

All authors

line 17, This study aimed to determine the cost-effectiveness of the HPV Obviously the authors meant vaccination against HPV.

Response: Many thanks for your valuable comment. The amendment was determined using green highlight [Abstract section, Line 18].

Line 191  Figure 2 shows the female mortality due to HPV

It is necessary to clarify whether we are talking about mortality in the country or per 100,000 population.

Response: The female mortality in the country. The amendment was determined using green highlight [Line 192] 

Line 206 and 214. The aggregated annual female mortality is shown in Figure 3.

       See note to line 191.

Response: The female mortality in the country. The amendment was determined using green highlight [Line 208] 

Line 240 and 244. Additionally, it shows the incremental costs, QALYs and deaths.

Response: Many thanks for your valuable comment. The amendments were determined using green highlight [Lines 241-243].

Reviewer 2 Report

Comments and Suggestions for Authors

The authors have conducted a comprehensive and insightful analysis on the cost-effectiveness of HPV vaccination strategies in Iran. Their work contributes valuable knowledge to the field of public health, especially in low- and middle-income countries. However, to further enhance the manuscript and its impact on the reader and the scientific community, here are some comments and suggestions:
- Broadening the Context
While the study focuses on Iran, discussing its implications for similar low- and middle-income countries could broaden its relevance. This could include a comparison with countries that have successfully implemented HPV vaccination programs and the lessons learned from those experiences.

- Addressing Limitations More Thoroughly
The authors briefly mention the limitations of their study, such as the conservative nature of their estimates and the specific focus on cervical cancer without considering other HPV-related cancers. Expanding on these limitations and discussing how future research could address them would strengthen the paper. This might include potential methodologies for incorporating the societal costs and benefits of vaccination against other types of HPV-related cancers.

- Clarification on Model Parameters
The manuscript could benefit from a more detailed explanation of the model parameters, especially those that are unique to the Iranian context or differ significantly from parameters used in similar studies. This would help readers better understand the assumptions underpinning the analysis and how they affect the results.

- Sensitivity Analysis
If not already included to a sufficient degree, conducting and presenting a comprehensive sensitivity analysis would be valuable. This analysis should explore how variations in key assumptions (e.g., vaccine cost, vaccine effectiveness over time, and coverage rates) impact the cost-effectiveness outcomes. This would provide a more robust understanding of the conditions under which vaccination strategies remain viable.

Comments on the Quality of English Language

The text is clear and concise, employing appropriate technical language for the field of public health and epidemiology. Specialized terms are used correctly, facilitating understanding for readers familiar with the subject.

Author Response

Dear reviewer,

Thank you for your comments. We have attempted to address them a complete and suitable as possible. Below your questions and our responses. In addition the adjustments in the new submitted document are marked green.

Kind regards,

All authors

  1. Broadening the Context

While the study focuses on Iran, discussing its implications for similar low- and middle-income countries could broaden its relevance. This could include a comparison with countries that have successfully implemented HPV vaccination programs and the lessons learned from those experiences.

Response: Many thanks for your valuable comment. The issue was added to discussion section using green highlight [lines 316-320].

a- Bangura MS, Zhao Y, Gonzalez Mendez MJ, Wang Y, Didier Sama S, Xu K, Ren R, Ma L, Qiao YL. Case study of cervical cancer prevention in two sub-Saharan African countries: Rwanda and Sierra Leone. Front Med (Lausanne). 2022 Sep 15;9:928685. doi: 10.3389/fmed.2022.928685. PMID: 36186799; PMCID: PMC9521665.

b-Sayinzoga F, Umulisa MC, Sibomana H, Tenet V, Baussano I, Clifford GM. Human papillomavirus vaccine coverage in Rwanda: A population-level analysis by birth cohort. Vaccine. 2020;38:4001-4005.

  1. Addressing Limitations More Thoroughly

The authors briefly mention the limitations of their study, such as the conservative nature of their estimates and the specific focus on cervical cancer without considering other HPV-related cancers. Expanding on these limitations and discussing how future research could address them would strengthen the paper. This might include potential methodologies for incorporating the societal costs and benefits of vaccination against other types of HPV-related cancers.

Response: We have included a extra bit in the discussion on the limitations and possible methodological suggestions to capture these.

  1. Clarification on Model Parameters

The manuscript could benefit from a more detailed explanation of the model parameters, especially those that are unique to the Iranian context or differ significantly from parameters used in similar studies. This would help readers better understand the assumptions underpinning the analysis and how they affect the results.

Response: We have included a section Material and Methods considering the parameters and the country differences. I hope you understand that we assumed certain epidemiological parameters,  as they are hard to find for Iran and often do not deviate considerably.

  1. Sensitivity Analysis

If not already included to a sufficient degree, conducting and presenting a comprehensive sensitivity analysis would be valuable. This analysis should explore how variations in key assumptions (e.g., vaccine cost, vaccine effectiveness over time, and coverage rates) impact the cost-effectiveness outcomes. This would provide a more robust understanding of the conditions under which vaccination strategies remain viable.

Response:

Thank you for your suggestion. We have now included a sensitivity analysis section in 3.2 Scenario analysis. A one-way sensitivity analysis was carried out on in the scenarios and also appropriate adjustments were made in the discussion and the methods. 

  1. Comments on the Quality of English Language

The text is clear and concise, employing appropriate technical language for the field of public health and epidemiology. Specialized terms are used correctly, facilitating understanding for readers familiar with the subject.

Response: Many thanks for the comment.